# Adaptive control for circulating cooling water system using deep reinforcement learning

Jin Xu[☯], Han Li[☯], Qingxin Zhang[ID]*[☯]

School of Artificial Intelligence, Shenyang Aerospace University, Liaoning, China

☯ These authors contributed equally to this work.
* zhy9712_sau@163.com

**Data Availability Statement:** All relevant data are within the manuscript and its Supporting Information files.

**Funding:** The author(s) received no specific funding for this work.

## Abstract

Due to the complex internal working process of circulating cooling water systems, most traditional control methods struggle to achieve stable and precise control. Therefore, this paper presents a novel adaptive control structure for the Twin Delayed Deep Deterministic Policy Gradient algorithm, which is based on a reference trajectory model (TD3-RTM). The structure is based on the Markov decision process of the recirculating cooling water system. Initially, the TD3 algorithm is employed to construct a deep reinforcement learning agent. Subsequently, a state space is selected, and a dense reward function is designed, considering the multivariable characteristics of the recirculating cooling water system. The agent updates its network based on different reward values obtained through interactions with the system, thereby gradually aligning the action values with the optimal policy. The TD3-RTM method introduces a reference trajectory model to accelerate the convergence speed of the agent and reduce oscillations and instability in the control system. Subsequently, simulation experiments were conducted in MATLAB/Simulink. The results show that compared to PID, fuzzy PID, DDPG and TD3, the TD3-RTM method improved the transient time in the flow loop by 6.09s, 5.29s, 0.57s, and 0.77s, respectively, and the Integral of Absolute Error(IAE) indexes decreased by 710.54, 335.1, 135.97, and 89.96, respectively, and the transient time in the temperature loop improved by 25.84s, 13.65s, 15.05s, and 0.81s, and the IAE metrics were reduced by 143.9, 59.13, 31.79, and 1.77, respectively. In addition, the overshooting of the TD3-RTM method in the flow loop was reduced by 17.64, 7.79, and 1.29 per cent, respectively, in comparison with the PID, the fuzzy PID, and the TD3.

## 1 Introduction

In the production process of many industrial sectors, a large amount of waste heat will be generated. Currently, it is necessary to use cold water or other liquids to absorb the heat in time to ensure the regular operation of the production process. The cold water used in this process is called cooling water in industrial production. To save water resources and reduce energy costs, consider recycling industrial cooling water to form a circulating cooling water system. With the continuous development of modern industry, as an essential cooling method in industrial production, circulating cooling water system is widely used in various production processes,

**Competing interests:** The authors have declared that no competing interests exist.

such as pharmaceutical, electric power, chemical, metallurgy, Marine engine, etc. Optimizing circulating cooling water system control can improve industrial production efficiency and reduce energy consumption and maintenance costs. Therefore, in controlling circulating cooling water systems, the study of achieving efficient control has become an important topic.

At present, the control methods employed in circulating cooling water systems are predominantly based on traditional PID control [1,2], fuzzy control [3–5] model predictive control (MPC) [6,7], intelligent optimization algorithms [8–10] and other traditional methods. For example, Xia et al. [11] proposed the use of a PID controller and a fuzzy PID controller as the control strategy for the temperature controller of the circulating cooling water system in a fuel cell engine. This approach resulted in a notable reduction in temperature fluctuations during the water temperature mixing process. Terzi et al. [12] proposed the use of a model predictive control algorithm as the control strategy for an industrial plant's circulating cooling water system, which resulted in an improvement in control performance. Zhang et al. [13] developed an algorithmic coupling of an artificial neural network optimized by a genetic algorithm and a heat transfer model of the condenser and air-cooling heat exchanger. This was employed to optimize and control the mass flow of circulating cooling water in an indirect cooling system of thermal power units. The objective was to enhance the efficiency of the circulating cooling water system and to reduce costs. However, these methods all have certain limitations, which are difficult to adapt to the nonlinear dynamic characteristics of the system and the uncertainty in the operation process. To a significant extent, these methods depend on prior knowledge and necessitate the development of sophisticated system models and the adjustment of parameters. For instance, the PID control method necessitates manual system adjustment and is unable to accommodate the intricate dynamic alterations of the system. Although the fuzzy control method can effectively handle uncertainty, it often requires considerable expertise to design fuzzy rules and may struggle to achieve optimal control. Conversely, MPC can optimize control strategies based on predictive control, thereby enhancing control performance. However, MPC is characterized by high computational complexity and demands significant computing resources, which presents a challenge in the application of real-time, high-frequency control systems. Furthermore, MPC is susceptible to model accuracy and measurement precision, which may result in suboptimal performance in cases of unknown systems or model errors. The intelligent optimization algorithm exhibits strong adaptability but may be prone to becoming stuck in a suboptimal local solution, thus failing to ensure optimal control of the system.

In recent years, with the continuous development of artificial intelligence technology, artificial intelligence theories and technologies such as deep learning and reinforcement learning have been widely applied in many fields, such as games field [14,15], robot control field [16–18], building energy efficiency field [19], natural language processing field [20], and automatic driving field [21–23], and fault diagnosis field [24,25]. RL is a machine learning method to learn the optimal decision through trial and error. It is powerful nonlinear modeling and adaptive learning ability have brought new opportunities for controlling the circulating cooling water system. For example, Qiu et al. [26] proposed a model-free optimal control method based on reinforcement learning to control circulating cooling water systems in the architectural field, which makes it have broad application prospects in the architectural area where accurate system performance models are generally lacking. Wu et al. [27] proposed a PI controller based on reinforcement learning to control a steam compression refrigeration system with nonlinearity and coupling two inputs and two outputs, realizing adaptive control and improving control performance. Compared with traditional control methods, the reinforcement learning method can automatically learn the system's dynamic characteristics and operation rules without manually adjusting the control parameters and has better adaptability and

intelligence. In addition, the reinforcement learning method can also use multi-agent reinforcement learning to realize collaborative control among multiple circulating cooling water systems and further improve control efficiency and stability. For example, Fu et al. [28] proposed a multi-agent deep reinforcement learning method for building cooling water system control to optimize the load distribution, cooling tower fan frequency, and cooling pump frequency of different cooling water systems. Furthermore, industrial processes' safety usually requires solving constrained optimal control (COC) problems. Zhang et al. [29] proposed a new safety-enhanced learning algorithm for COC problems of continuous-time nonlinear systems with unknown dynamics and perturbations. For the uncertainties in the bridge crane system, such as payload mass and unmodeled dynamics, without knowing the system model, a new model-free online reinforcement learning control method for the real-time position adjustment and anti-sway control problem of bridge cranes is proposed [30], which combines the advantages of adaptive and optimal control and exhibits satisfactory performance. These research results show that reinforcement learning methods have a broad application prospect in industrial process control.

In order to ascertain whether deep reinforcement learning methods offer certain advantages over traditional control methods in the recirculating cooling water system, and to address issues such as the inability of traditional control methods to achieve stable and precise control of the controlled system, this paper proposes the design of an adaptive control structure for the recirculating cooling water system with the objective of improving the system's control performance. This paper makes the following contributions:

1) The design of an adaptive control structure based on the Twin Delayed Deep Deterministic Policy Gradient algorithm under a reference trajectory model (TD3-RTM) enables end-to-end control of the recirculating cooling water system at the simulation level.

2) The state space and reward function were designed to consider the multivariable characteristics of the recirculating cooling water system. A reference trajectory model was introduced to accelerate the convergence speed of the agent and reduce oscillations and instability in the control system.

3) The exploration of the potential application of deep reinforcement learning in the recirculating cooling water system, with the objective of providing references and insights for control problems in the industrial field.

The rest of this paper is organized as follows: Section 2 is the background, introducing the basics of deep reinforcement learning, the working principle of the circulating cooling water system, and the system model. Section 3 is the methods, which Outlines the design of adaptive control structure based on TD3-RTM. Section 4 is the experiment and analysis of results. Section 5 is the conclusion, which summarizes this study and puts forward the future research direction.

## 2 Background

The prerequisite for combining the control of a circulating cooling water system with reinforcement learning is establishing a Markov model of the circulating cooling water system. The working principle and model of the circulating cooling water system and Markov decision process (MDP) based on the circulating cooling water system are described below.

### 2.1 Circulating cooling water system

The circulating cooling water system comprises a temperature sensor, flowmeter, pressure sensor, heat exchanger, electric control valve, manual butterfly valve, check valve, frequency conversion pump, and other equipment. The schematic system diagram is shown in Fig 1.

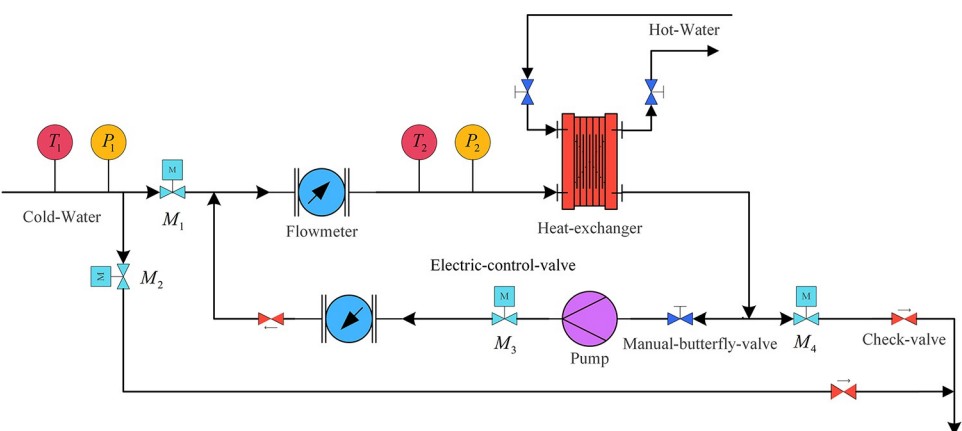

**Fig 1. Schematic diagram of circulating cooling water system.**

The cold-water flow into the line through the electric regulating valve $M_1$. When the pressure sensor $P_1$ detects that the pressure in the main line of the system exceeds the safety value required by the system, the opening of the electric regulating valve $M_2$ increases and discharges part of the cold water for pressure relief. At the same time, $M_2$ is connected to the check valve to prevent backflow. Another amount of cold water enters the main line through $M_1$, is detected by the flow meter and temperature sensor $T_2$, and then enters the heat exchanger. As the heat exchange proceeds, some hot water is discharged through an electrically regulated valve $M_4$ connected to a check valve. Another part of the hot water is mixed with the cold water through an electrically controlled valve $M_3$. This cycle ensures that the cold water flowing into the heat exchanger has a constant temperature, thus ensuring the stability and safety of the system.

Flow and temperature are crucial control objectives in a circulating cooling water system. To simplify modeling and control complexity, this paper focuses on these two critical variables as the primary targets for controlling the circulating cooling water system. On the other hand, over the past decades, the successful application of single-variable control theory has demonstrated the convenience and effectiveness of using transfer functions to express and analyze control systems. Therefore, transfer function matrices are employed in this paper to describe and analyze circulating cooling water systems.

This paper represents the circulating cooling water system as a multivariate model with two inputs and outputs, as shown in Fig 2. The input variables are the opening of electric control valve $M_1$ and $M_3$. The output variables are the water flow and temperature into the heat exchanger. The linear transfer functions $G_{11}$, $G_{12}$, $G_{21}$, and $G_{22}$ represent the relationship between the input and output variables of the system, where the first number represents the output, and the second number represents the input. For example, $G_{21}$ represents the effect of valve opening $M_1$ on temperature.

The transfer functions $G_{11}$, $G_{12}$, $G_{21}$, and $G_{22}$, which represent the dynamic behavior of the system, need to be identified, and the best pairing variables need to be found for the controller design. Therefore, the best-paired variables are found by selecting different variable pairs to observe the regulation state of the system during the experiment and then collecting data on the input and output quantities of the system at a steady state. For the collected system data, the data is firstly pre-processed to remove the outliers and noise. Then the transfer function model $G(S)$ of the circulating cooling water system [31] is obtained using the MATLAB system

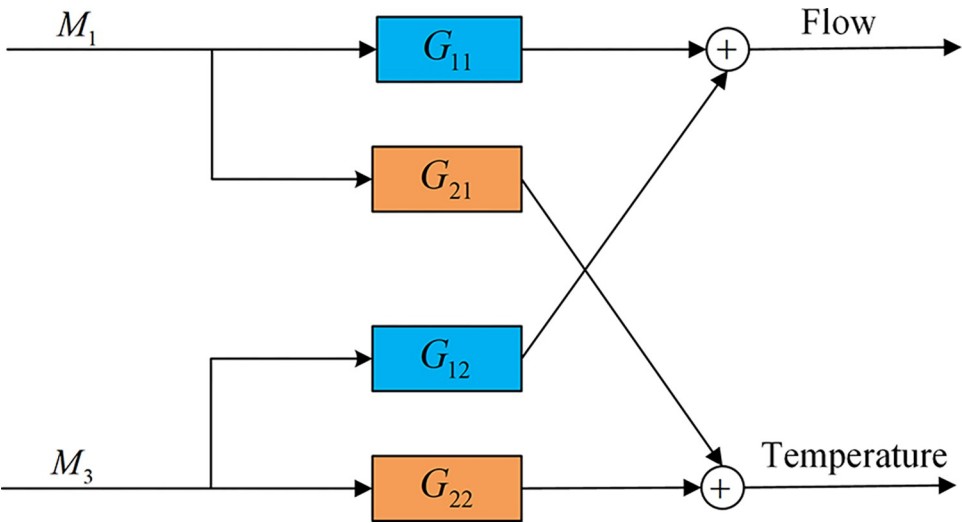

**Fig 2. Model of circulating cooling water system.**

identification toolbox, as shown in Eq 1.

$$G(S) = \begin{bmatrix} \dfrac{0.7541S + 0.002914}{S^2 + 0.08358S + 0.0002578} & \dfrac{24.65S + 0.02572}{S^2 + 2.529S + 0.003538} \\ \dfrac{4.721e - 05S - 3.809e - 06}{S^2 + 0.2304S + 4.309e - 14} & \dfrac{0.354S + 0.0006877}{S^2 + 1.189S + 0.002565} \end{bmatrix} \quad (1)$$

## 2.2 Markov decision model of circulating cooling water system

The mathematical foundation and modeling tool of reinforcement learning is the MDP. An MDP usually comprises state space $s$, action space $a$, state transition function $P$, reward function $r$, and discount factor $\gamma$. At any time step $t$, the agent first observes the current state $s_t$ of the environment and the current corresponding reward value $r_t$. Based on this state and reward information, the agent acts $a_t$ and obtains the state $s_{t+1}$ and the reward $r_{t+1}$ from the environment for the next step. The interaction between the reinforcement learning agent and the environment under the control system is shown in Fig 3.

In control system terminology, the term "agent" refers to the designed controller; the "environment" includes the system outside the controller, which, in this paper, refers explicitly to the circulating cooling water system. The policy represents the optimal control behavior sought by the designer. As shown in Fig 3, the interaction process between the agent and the environment indicates that state $s$ represents various features and parameters measured by sensors in the circulating cooling water system, such as flow and temperature. Action $a$ illustrates the opening value of the electric regulating valve determined by the agent based on the current state of the circulating cooling water system. Reward $r$ indicates the feedback obtained by the agent after taking specific actions in specific conditions. Rewards are used to evaluate the quality of the agent's behavior and guide decision-making in different states. In the context of the circulating cooling water system, rewards can be used to measure the control effectiveness and performance of the system. In deep reinforcement learning, state transition function $P$ is often unknown, so the agent needs to estimate the state transition probability through interaction with the environment and learn and optimize control strategies. The design of the

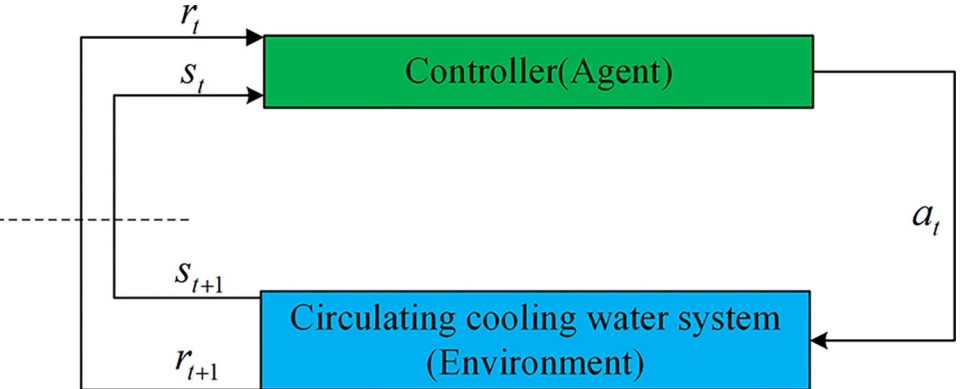

**Fig 3. Agent and environment interaction process.**

control strategy based on deep reinforcement learning relies on the design of the state, action, reward function, and reinforcement learning algorithms. Section 3 will provide a detailed introduction to the design of the control strategy.

## 3 Methods

In this study, a deep reinforcement learning approach is used to design an adaptive controller for the circulating cooling water system. In deep reinforcement learning, the neural network is used as the value function or parameterized policy, while the gradient optimization method is used to optimize the loss. Here, the twin delayed deep deterministic policy gradient [32] (TD3) algorithm, which is an actor-critic framework to deal with continuous action space problems, is employed to optimize the control parameters in the circulating cooling water system.

The TD3 algorithm is a deep reinforcement learning algorithm based on the Actor-Critic framework based on the Deep Deterministic Policy Gradient [33] (DDPG) algorithm. Since the value network of DDPG tends to overestimate the action value function, the TD3 algorithm has made improvements in the following three aspects to address the shortcomings of the DDPG algorithm: with truncated double Q learning, the problem of overestimation of the critic network is alleviated; the robustness and smoothness of the algorithm are improved by adding noise that obeys a truncated normal distribution to the output action of the target policy network; make the policy network and the three target networks update less frequently than the value network, this method can reduce the variance of the approximate action-value function, and a better policy can be obtained. The TD3 algorithm was selected for controlling the circulating cooling water system in this study due to its capability to handle continuous action spaces, utilization of twin Q networks to mitigate overestimation bias, and implementation of delayed policy updates and soft updates to reduce function approximation errors, thereby delivering more stable and precise control. Furthermore, TD3's deep neural networks are proficient in effectively modeling the complex nonlinear and multivariate characteristics of the system, facilitating real-time adaptation and optimized control, thereby enhancing system performance.

Choosing an appropriate deep reinforcement learning algorithm is only part of designing a controller. In contrast, the design of states, actions, and rewards in reinforcement learning are crucial in determining the agent's learning capabilities, control performance, and adaptability to dynamic environments. Thoughtful and well-tailored designs of these elements are essential for successful and efficient learning in various applications. The following will explain the selection and design of states, actions, and reward functions.

### 3.1 Control strategy design

**3.1.1 State.** The state reflects essential information during the interaction between the agent and the environment, and the selection of the state space directly affects the agent's decision-making, thereby influencing the overall control performance of the system. Therefore, the state should contain sufficient information to describe the current stage. In the circulating cooling water system, where the actuator exhibits nonlinear characteristics, and the process gain varies with different manipulated variables, the chosen state space in this study is as follows:

$$s = [eF, eT, \int eFdt, \int eTdt, F, T, Fsp, Tsp, a_1, a_2]^{\mathrm{T}} \tag{2}$$

Where, $eF = F_{sp}-F$ and $eT = T_{sp}-T$ represent the control error values of the flow and temperature loops, respectively. $\int eFdt$ and $\int eTdt$ are the error integrals. $F$ and $T$ represent the historical output measurement values of flow and temperature, respectively. $Fsp$ and $Tsp$ are the setpoints for flow and temperature. $a_1$ and $a_2$ are the manipulated variable values for the flow and temperature loops, respectively, which are the action values output by the agent.

**3.1.2 Action.** Actions represent the actions taken by the agent in specific states, and the agent's task is to choose appropriate actions in different states to maximize its long-term rewards. In reinforcement learning, actions are typically determined by the agent's policy, and in control systems, they correspond to the manipulated variables applied to the system. In this study, the action values correspond to the opening values of the electric regulating valves in the circulating cooling water system, where $a_1$ represents the opening value of the electric regulating valve $M_1$, and $a_2$ represents the opening value of the electric regulating valve $M_3$. The range of action values is [0, 100], making the action space:

$$a = [a_1, a_2]^{\mathrm{T}} \tag{3}$$

**3.1.3 Reward.** The reward function is a crucial concept in reinforcement learning, which is used to evaluate the performance of an agent in an environment. The reward function is typically a mapping from the state and action space to a real number, representing the desirability of an action taken by the agent in each state. In reinforcement learning, the objective of the agent is to maximize the accumulated reward by interacting with the environment. Therefore, the reward function can be viewed as the objective function of the reinforcement learning task. By adjusting its policy, the agent can attempt to maximize the reward function and learn how to take optimal actions in different states of the environment.

In some reinforcement learning tasks, the reward function is typically designed such that the agent receives a reward only when the output values satisfy the system requirements. This type of reward function is known as a sparse reward function. In simple environments like single-variable systems, using a sparse reward function can still yield good control results. However, in a multivariate system, transferring the state of the system environment to the target state becomes more complex and uncertain than that of a univariate system. Therefore, based on the characteristics of circulating cooling water systems, this paper designs a dense reward function. For the flow loop, the dense reward function is set as follows:

$$r_1 = \begin{cases} 100, & |eF_t| \leq \varphi_1 \\ 1/eF_t, & \varphi_1 < |eF_t| \leq \varphi_2 \\ -eF_t, & |eF_t| > \varphi_2 \end{cases} \tag{4}$$

Where $eF_t$ represents the error value of the current moment of flow. $\varphi_1$ and $\varphi_2$ represent the thresholds of error values in different intervals of flow. When the error value satisfies the system goal requirements, give the agent a large reward value to encourage the current behavior. In this paper, $\varphi_1$ equals 0.1, and $\varphi_2$ equals 5.

Furthermore, the temperature loop reward function is designed in the same way as the flow loop. Therefore, the reward function of the temperature loop is defined as:

$$r_2 = \begin{cases} 100, & |eT_t| \leq \eta_1 \\ 1/eT_t, & \eta_1 < |eT_t| \leq \eta_2 \\ -eT_t, & |eT_t| > \eta_2 \end{cases} \tag{5}$$

Where $eT_t$ represents the error value of the current moment of temperature. $\eta_1$ and $\eta_2$ represent the thresholds of error values in different intervals of temperature. In this paper, $\eta_1$ equals 0.1, and $\eta_2$ equals 2.

Finally, the reward function $r_t$ based on the circulating cooling water system is defined as

$$r_t = r_1 + r_2 \tag{6}$$

## 3.2 Network structure and algorithm design

The network structure of the TD3 algorithm comprises four principal components: the Actor network, the Critic network, the Target Actor network, and the Target Critic network. The Actor network generates a policy for continuous actions based on the current state. The Critic network is responsible for estimating the Q-value for the current state and action pair. The Target Actor and Target Critic networks serve as target networks for the Actor and Critic networks, respectively. The Target Actor and Target Critic networks have the same structure as the Actor and Critic networks, respectively, and their parameters are updated through soft updates from the Actor and Critic networks. In this study, the Actor and Critic networks are implemented with three-layer neural networks, comprising 128 and 64 neurons in their respective hidden layers. The rectified linear unit (ReLU) function is employed as the activation function. Furthermore, as the control actuator in the circulating cooling water system is an electric regulating valve with a range of 0 to 100, the output of the Actor network is normalized to the range of [−1, 1] using the tanh function and then scaled using the scaling operation.

To enhance the exploration and learning capabilities of the agent, this paper introduces a reference trajectory model. This model guides the agent to converge more rapidly to the desired control policy during the learning process, thereby improving the control effectiveness and learning speed of reinforcement learning. The reference trajectory model utilized in this study is:

$$F_r(s) = \frac{1}{\tau_r s + 1} \tag{7}$$

In addition, in practical applications, setpoints may experience sudden changes or instability, which can lead to unstable performance or oscillations in the control system. By introducing the reference trajectory model, the setpoint signal can be smoothed to make its changes more gradual and smoother, thereby helping to reduce oscillations and instability in the control system. In this paper, $\tau_r$ equals 0.2. The design of the control system is illustrated in Fig 4.

In this study, a controller for the circulating cooling water system is designed based on the TD3 algorithm. Its control strategy is shown in Algorithm 1.

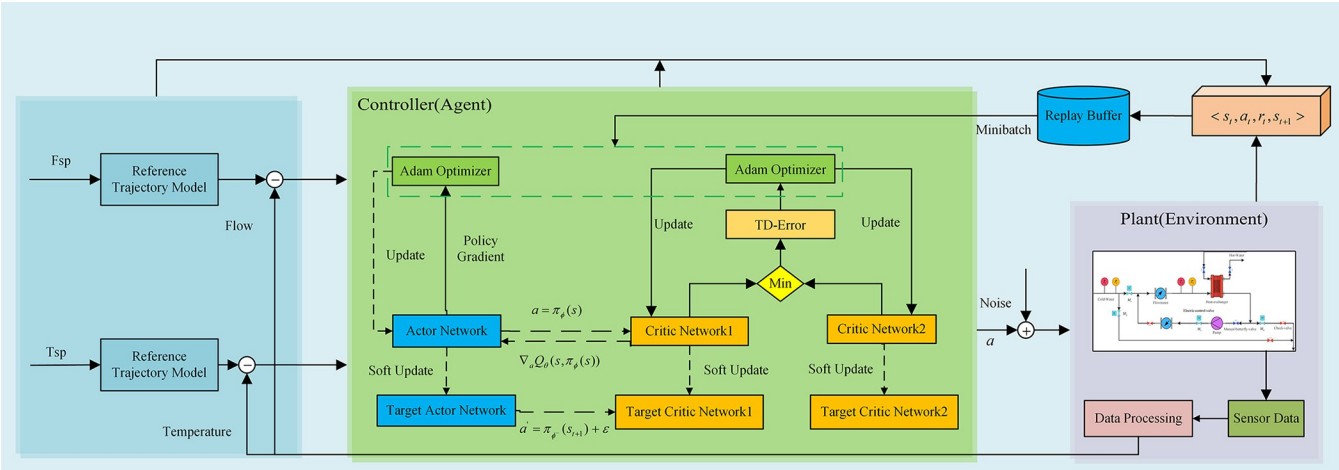

**Fig 4. Control structure of circulating cooling water based on TD3-RTM.**

**Algorithm 1. TD3 algorithm in circulating cooling water control system.**

Initialize replay buffer $M$, initialize critic network $Q_{\theta_1}, Q_{\theta_2}$, parameters $\theta_1$, $\theta_2$, initialize actor network $\pi_\phi$ parameter $\phi$, initialize target network parameters $\theta_1^- \leftarrow \theta_1, \theta_2^- \leftarrow \theta_2, \phi^- \leftarrow \phi$.
Repeat
  Randomly initialize the flow and temperature setpoints within the range allowed by system.
  select action $a_t \sim \pi_\phi(s_t) + \varepsilon, \varepsilon \sim N(0,\sigma)$, accept reward $r_t$ and next state $s_{t+1}$.
  store state transfer data ($s_t$, $a_t$, $r_t$, $s_{t+1}$) to $M$.
  Sample mini batches of size $B$ from $M$.
  $a'_{t+1} \leftarrow \pi_{\phi^-}(s_{t+1}) + \varepsilon, \varepsilon \sim \text{clip}(N(0,\sigma^-), -c, c), c > 0)$ $y = r_t + \gamma \min_{i=1,2} Q_{\theta_i^-}(s_{t+1}, a'_{t+1})$ Update value network.
  $\theta_i \leftarrow \text{argmin}_{\theta_i} B^{-1} \sum (y - Q_{\theta_i}(s_t, a_t))^2$ if $t$ mod $d$ then
    Update $\phi$
    $\nabla_\phi J(\phi) = B^{-1} \sum \nabla_a Q_{\theta_1}(s_t, a_t)|_{a_t = \pi_\phi(s_t)} \nabla_\phi \pi_\phi(s_t)$ Update the target network, where $\rho$ is the soft update factor.
    $\theta_i^- \leftarrow \rho \theta_i + (1 - \rho) \theta_i^-$
    $\phi^- \leftarrow \rho \phi + (1 - \rho) \phi^-$
  end if
end for

## 4 Experiments and analysis of results

In the training process of TD3-RTM in this study, the total number of episodes is set to 2000, with a sampling time of 0.1 seconds and a maximum simulation duration of 20 seconds. To enhance the disturbance rejection control performance of the system, random step signals with amplitudes ranging from -5 to 5 are applied at the control ports of the flow and temperature loops at the 15th second. The reference step input signals for the flow (m^3/h) and temperature (°C) are set to [550, 650] and [20, 30], respectively, to achieve robustness to significant setpoint changes in the system. Since TD3-RTM is based on the TD3 algorithm, the primary hyperparameters used in the training process of the TD3 algorithm are shown in Table 1.

**Table 1. Hyperparameter settings of the algorithm 1.**

| Hyperparameters | Values |
|---|---|
| Discount factor, $\gamma$ | 0.995 |
| Mini-batch size | 128 |
| Replay buffer size | 1e6 |
| Critic learning rate | 1e-3 |
| Actor learning rate | 5e-4 |
| Target update frequency | 10 |
| Exploration model | Gaussian noise |
| Variance, $\sigma$ | 0.2 |
| Variance decay rate | 1e-5 |
| Policy update frequency | 2 |
| Soft update factor, $\rho$ | 5e-3 |

All computations were carried out on a standard PC (Win11, AMD 4600H CPU@3.00GHz, 16GB) in MATLAB/Simulink R2022b. To validate the effectiveness of TD3-RTM, comparisons were made with the classical PID controller, fuzzy PID controller, DDPG algorithm, and TD3 algorithm. To be fair, the PID parameters for classical PID control and fuzzy PID control were obtained using the Ziegler-Nichols method. The neural network architecture, number of neurons, and learning rate used in different deep reinforcement learning algorithms were the same. Each task was run for 2000 episodes, and the experiments were repeated five times with different random seeds. The recorded results represent the average reward value for every 20 episodes. The learning curves are shown in Fig 5.

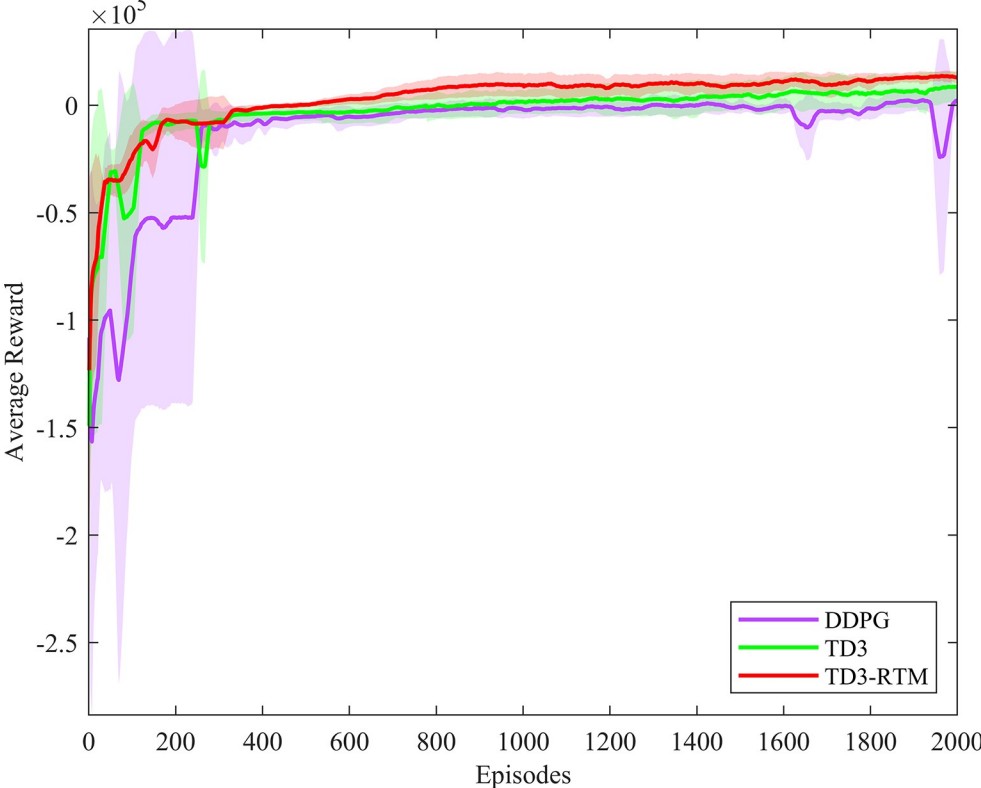

**Fig 5. Learning curve of the control task.**

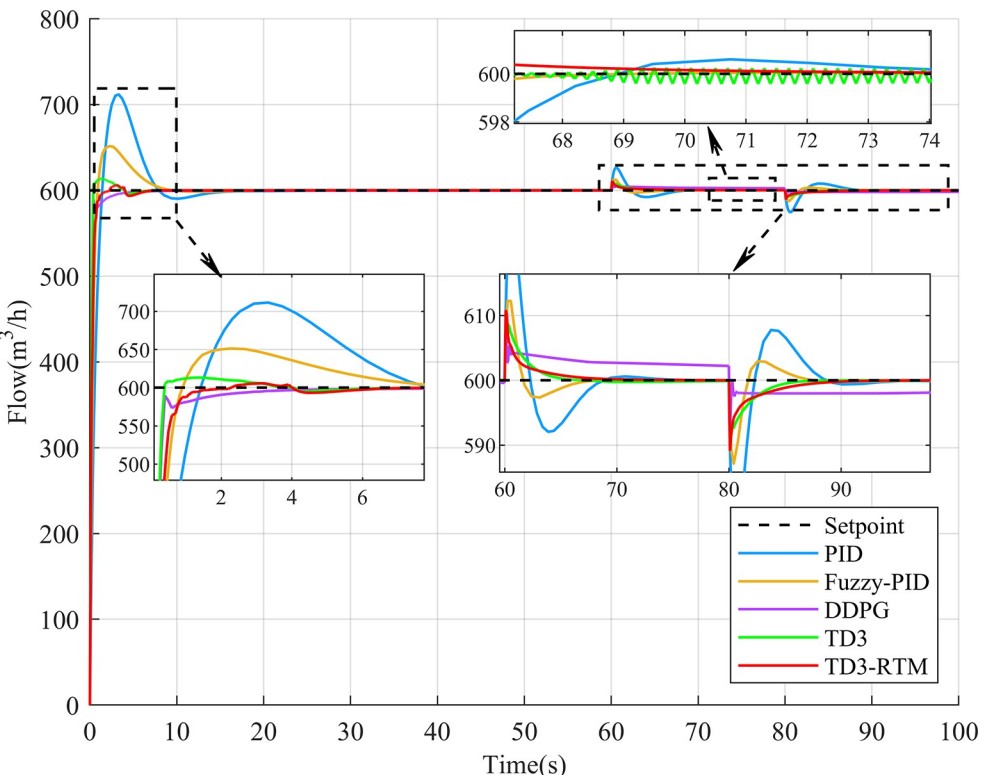

**Fig 6. Output results of different controllers at the flow setpoint of 600 m^3/h.**

The results in Fig 6 indicate that TD3-RTM can converge to the desired control policy faster and with more stable convergence performance under different random initial states. Additionally, TD3-RTM achieves higher total rewards after 2000 episodes of learning.

## 4.1 Step response and disturbance rejection performance simulation experiment

To validate the control effectiveness of TD3-RTM in the circulating cooling water system, a 100-second simulation experiment was conducted with a flow of 600 m^3/h and a temperature of 25˚C. At the 60-second and 80-second marks, disturbance signals with amplitudes of 5 and -5 were applied to the control ports of the flow and temperature loops. The control performance of different control methods is shown in Figs 6 and 7.

As shown in Fig 6, it can be observed that in the flow control loop, the deep reinforcement learning controller exhibits faster response speed and minor overshoot compared to the classical PID controller and fuzzy PID controller in the step response. TD3-RTM is less affected when faced with external disturbance signals, while the DDPG algorithm shows steady-state error and the TD3 algorithm exhibits oscillations. The oscillations in the TD3 algorithm cause continuous changes in the control signal, which can damage the actuators in the circulating cooling water system and lead to system instability. As shown in Fig 7, in the temperature control loop, the PID controller, fuzzy PID controller and DDPG algorithm have relatively slow response speeds, while the TD3 algorithm and TD3-RTM achieve good control performance. The deep reinforcement learning controller performs better when subjected to external disturbance signals. The performance parameters for different control methods are shown in Table 2.

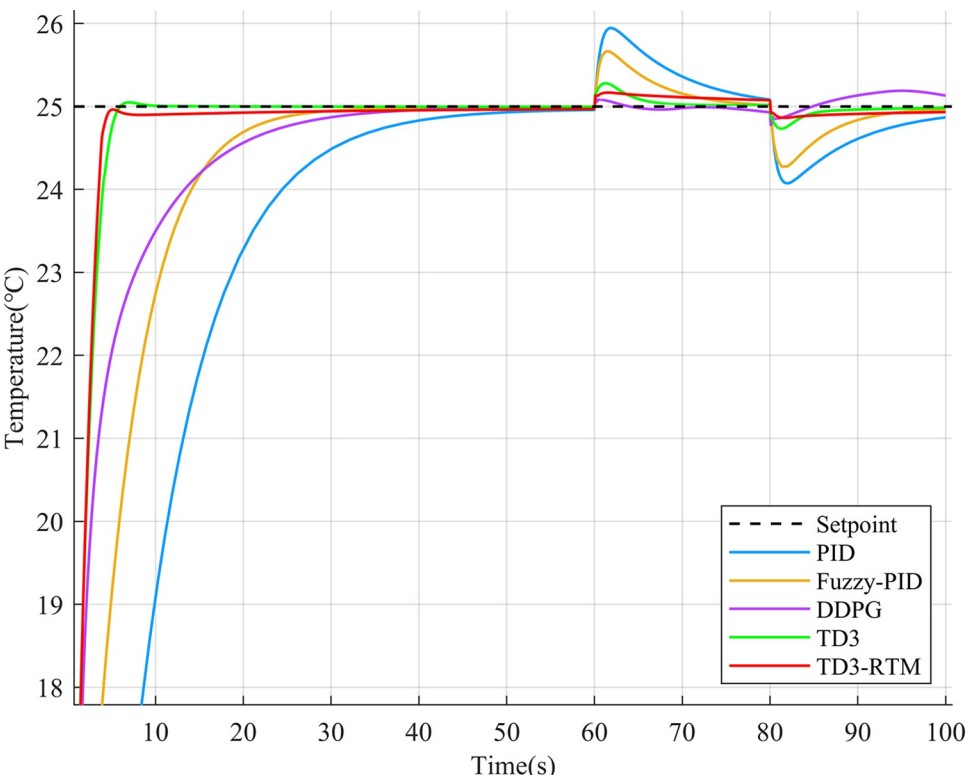

**Fig 7. Output results of different controllers at the temperature setpoint of 25˚C.**

From Table 2, compared to PID, fuzzy PID, DDPG and TD3, the TD3-RTM method improved the transient time in the flow loop by 6.09s, 5.29s, 0.57s, and 0.77s, respectively, and the Integral of Absolute Error(IAE) indexes decreased by 710.54, 335.1, 135.97, and 89.96, respectively, and the transient time in the temperature loop improved by 25.84s, 13.65s, 15.05s, and 0.81s, and the IAE metrics were reduced by 143.9, 59.13, 31.79, and 1.77, respectively. In addition, the overshooting of the TD3-RTM method in the flow loop was reduced by 17.64, 7.79, and 1.29 per cent, respectively, in comparison with the PID, the fuzzy PID, and the TD3. Generally, for industrial energy-consuming scenarios such as the circulating cooling water system, a controller with lower IAE and shorter settling time can save more energy. Although the DDPG algorithm shows a shorter rise time and no overshoot in the flow control

**Table 2. Comparison of controllers performance parameters.**

| Variables | Controllers | Rise Time (s) | Transient Time (s) | Overshoot (%) | IAE |
|---|---|---|---|---|---|
| Flow | PID | 1.03 | 7.10 | 18.60 | 753.80 |
| | Fuzzy-PID | 0.57 | 6.30 | 8.75 | 378.36 |
| | DDPG | 0.30 | 1.58 | 0 | 179.23 |
| | TD3 | 0.28 | 1.78 | 2.25 | 133.22 |
| | TD3-RTM | 0.47 | 1.01 | 0.96 | 43.26 |
| Temperature | PID | 16.54 | 29.81 | 0 | 170.35 |
| | Fuzzy-PID | 9.32 | 17.44 | 0 | 85.28 |
| | DDPG | 5.99 | 18.84 | 0 | 58.24 |
| | TD3 | 2.97 | 4.60 | 0.21 | 28.22 |
| | TD3-RTM | 2.78 | 3.79 | 0 | 26.45 |

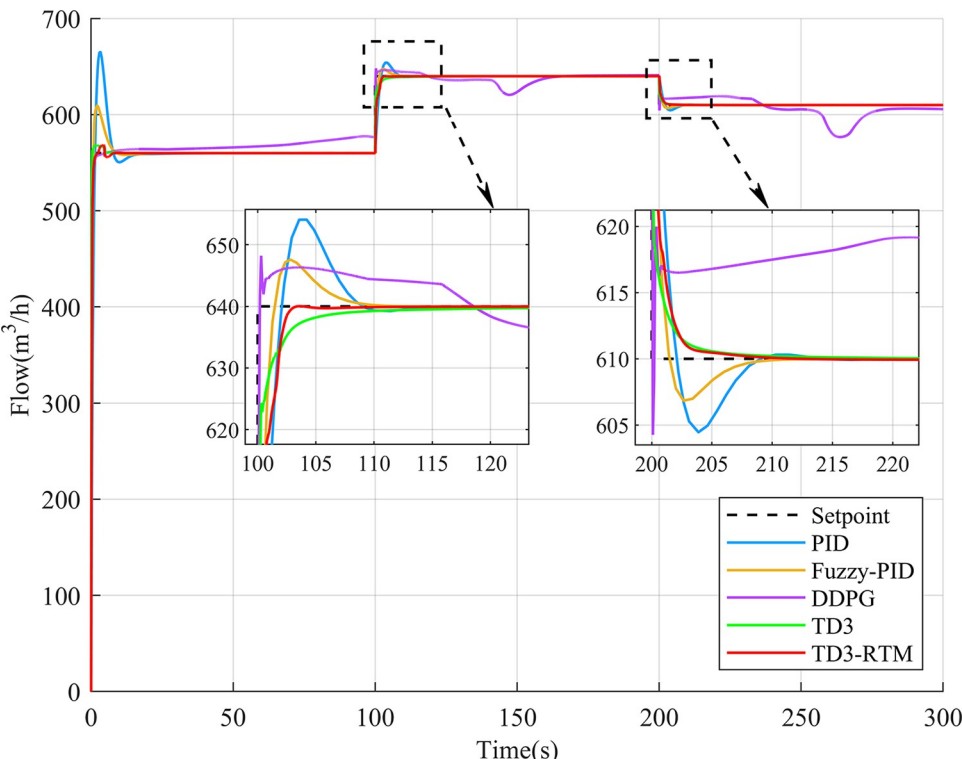

**Fig 8. Output results for different controllers in the flow loop at different setpoints.**

loop, it exhibits a longer rise time and settling time in the temperature control loop. Its IAE is the largest compared to TD3 and TD3-RTM. Overall, TD3-RTM demonstrates significant advantages in both control loops.

## 4.2 Tracking performance simulation experiment

To validate the tracking performance of TD3-RTM, this study designed different setpoints for both the flow control loop and the temperature control loop and conducted 300 seconds of simulation experiments. The control effects of different control methods are shown in Figs 8 and 9.

As shown in Fig 8, in the flow control loop, when the setpoint is changed, the DDPG algorithm becomes unstable, and both the PID controller and fuzzy PID controller exhibit significant overshoot and long settling times at different setpoints. On the other hand, the TD3 algorithm and TD3-RTM outperform other methods significantly. In Fig 9, although all controllers can track the given setpoints, their transient responses vary widely. The DDPG algorithm can reach the setpoint at the end of each simulation time, but its settling time is longer, and its performance is inferior to the PID and fuzzy PID controllers. However, TD3-RTM's performance during setpoint changes is comparable to the TD3 algorithm, with the fastest response speed and good settling time. Overall, TD3-RTM performs well in both control loops and shows excellent potential.

## 5 Conclusion

This paper presents a novel adaptive control structure based on the Twin Delayed Deep Deterministic Policy Gradient algorithm under a reference trajectory model (TD3-RTM) for

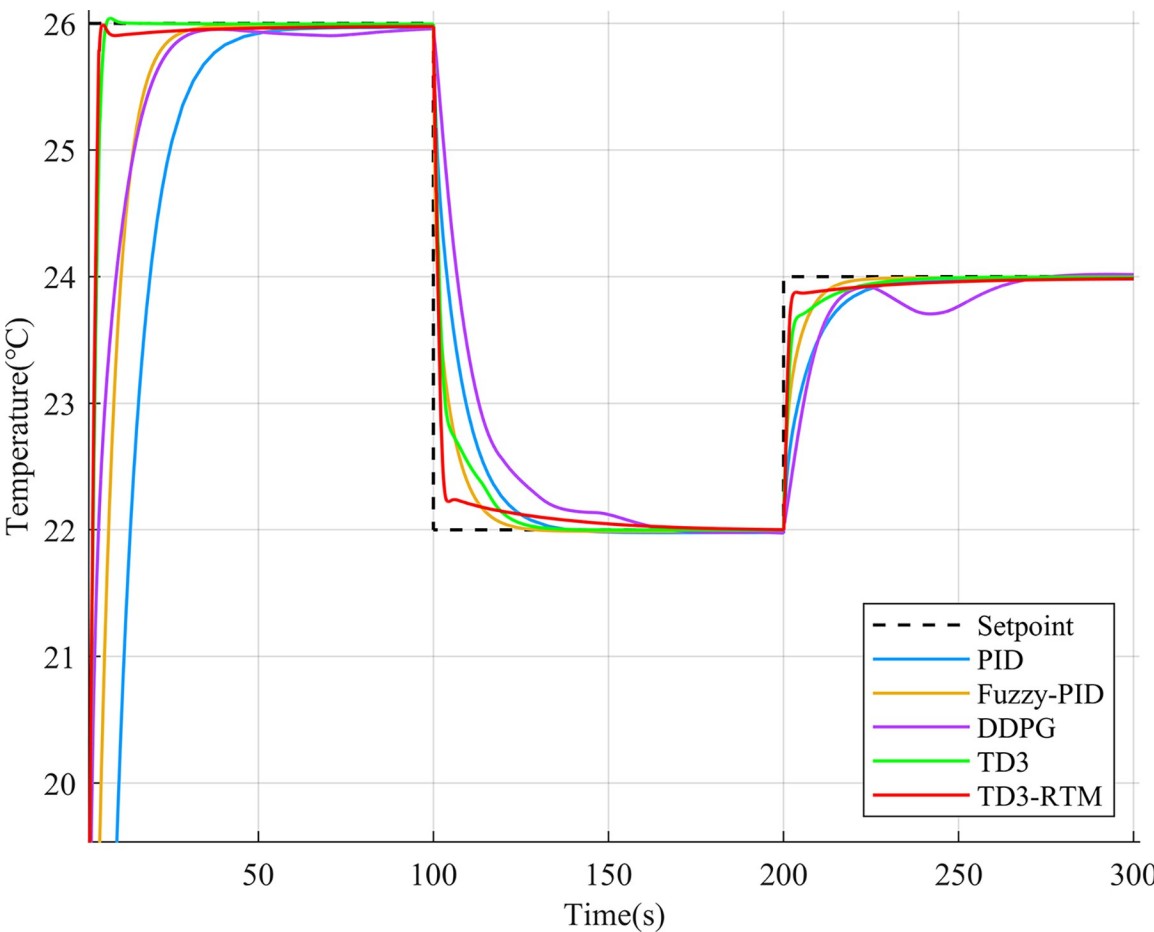

**Fig 9. Output results for different controllers in the temperature loop at different setpoints.**

addressing complex control problems in recirculating cooling water systems. Initially, the TD3 algorithm is employed to construct a deep reinforcement learning agent, enabling it to select appropriate actions for each loop based on system state features. Additionally, the multivariable characteristics of the recirculating cooling water system necessitate the design of a dense reward function, which enables the agent to receive various rewards through interactions with the environment and update its network, thereby gradually approaching the optimal policy. Furthermore, the introduction of the reference trajectory model accelerates the convergence speed of the agent and reduces system oscillations and instability. Simulation results show that compared to PID, fuzzy PID, DDPG and TD3, the TD3-RTM method improved the transient time in the flow loop by 6.09s, 5.29s, 0.57s, and 0.77s, respectively, and the Integral of Absolute Error(IAE) indexes decreased by 710.54, 335.1, 135.97, and 89.96, respectively, and the transient time in the temperature loop improved by 25.84s, 13.65s, 15.05s, and 0.81s, and the IAE metrics were reduced by 143.9, 59.13, 31.79, and 1.77, respectively. In addition, the overshooting of the TD3-RTM method in the flow loop was reduced by 17.64, 7.79, and 1.29 per cent, respectively, in comparison with the PID, the fuzzy PID, and the TD3. To further enhance safety and system stability, regular monitoring of system performance and adjusting as necessary are encouraged in practical applications. Furthermore, the utilization of backup control strategies to address exceptional circumstances that may be beyond the scope of deep reinforcement learning algorithms ensures the system's stability in extreme conditions.

This research validates the potential of deep reinforcement learning in the circulating cooling water system and offers novel solutions and insights for practical engineering control problems. Although the method proposed in this paper achieves good control performance in simulation experiments and shows advantages over both traditional control methods and other deep reinforcement learning methods, there are some potential limitations, such as applicability limitations, computational resource requirements, hyper-parameter sensitivity, adaptability to environmental variations, and the challenge of practical system validation. Further optimization and extension of the proposed control method can be explored for broader industrial applications, along with investigating other deep reinforcement learning algorithms for complex system control. This will contribute to advancing intelligent control technology in industrial automation, enhancing production efficiency and resource utilization.

## Supporting information

**S1 Appendix. Datas and codes from the experiments.**
(ZIP)

## Author Contributions

**Conceptualization:** Jin Xu.

**Data curation:** Han Li.

**Formal analysis:** Jin Xu.

**Methodology:** Jin Xu.

**Resources:** Qingxin Zhang.

**Software:** Han Li.

**Supervision:** Qingxin Zhang.

**Visualization:** Han Li.

**Writing – original draft:** Han Li.

**Writing – review & editing:** Qingxin Zhang.

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
