## [Decision Letter · Decision Letter 0]

19 Oct 2023

PONE-D-23-24165Adaptive control for circulating cooling water system using deep reinforcement learningPLOS ONE

Dear Dr. Zhang,

Thank you for submitting your manuscript to PLOS ONE. After careful consideration, we feel that it has merit but does not fully meet PLOS ONE’s publication criteria as it currently stands. Therefore, we invite you to submit a revised version of the manuscript that addresses the points raised during the review process.

We look forward to receiving your revised manuscript.

Kind regards,

Prof. Dr. Stelios Bekiros, PhD

Academic Editor

PLOS ONE

Journal Requirements:

Additional Editor Comments :

REVIEWER COMMENTS:

This paper presents a deep RL-based control of a circulating cooling water system. The topic has some practical significance, but the novelty of the paper is not enough. However, the decision can be reconsidered if the authors could carefully address all the concerns raised.

1. How does the proposed method ensure the stability of the system?

2. The authors mentioned many successful applications of RL to circulating cooling water system ([26-28]), what is the contribution of this manuscript compared to them? It is suggested that the motivation and contributions should be more emphasized.

3. Since there are many related methods that can also deal with optimal control of unknown systems, it is better to provide a more comprehensive literature review. Please note that the up-to-date of references will contribute to the up-to-date of your manuscript. The studies named: Robust safe reinforcement learning control of unknown continuous-time nonlinear systems with state constraints and disturbances, Journal of Process Control; Online reinforcement learning with passivity-based stabilizing term for real time overhead crane control without knowledge of the system model, Control Engineering Practice, can be used to explain the method in the study or to indicate the contribution in the "Introduction" section. I believe this would further strengthen the introduction and lend support to the methodology used in general.

4. Check the notation system throughout the text. For example, the differential operator in equation (1) and the state in MDP use the same character "s". The transfer function G and the state transition function P should be unified, the current expression is confusing. If a1 and M1 represent the same value, why do the authors use different notations?

5. The control error values in equation (2) are not defined. The error between what? It is suggested that the reference trajectory model be placed in a more appropriate location.

6. What is the difference between the proposed method and TD3?

7. Please improve the quality of all figures and the language.

Reviewers' comments:

Reviewer's Responses to Questions

**Comments to the Author**

1. Is the manuscript technically sound, and do the data support the conclusions?

Reviewer #1: Yes

2. Has the statistical analysis been performed appropriately and rigorously? 

Reviewer #1: Yes

3. Have the authors made all data underlying the findings in their manuscript fully available?

Reviewer #1: Yes

4. Is the manuscript presented in an intelligible fashion and written in standard English?

Reviewer #1: Yes

5. Review Comments to the Author

Reviewer #1: This paper presents a deep RL-based control of a circulating cooling water system. The topic has some practical significance, but the novelty of the paper is not enough. However, the decision can be reconsidered if the authors could carefully address all the concerns raised.

1. How does the proposed method ensure the stability of the system?

2. The authors mentioned many successful applications of RL to circulating cooling water system ([26-28]), what is the contribution of this manuscript compared to them? It is suggested that the motivation and contributions should be more emphasized.

3. Since there are many related methods that can also deal with optimal control of unknown systems, it is better to provide a more comprehensive literature review. Please note that the up-to-date of references will contribute to the up-to-date of your manuscript. The studies named: Robust safe reinforcement learning control of unknown continuous-time nonlinear systems with state constraints and disturbances, Journal of Process Control; Online reinforcement learning with passivity-based stabilizing term for real time overhead crane control without knowledge of the system model, Control Engineering Practice, can be used to explain the method in the study or to indicate the contribution in the "Introduction" section. I believe this would further strengthen the introduction and lend support to the methodology used in general.

4. Check the notation system throughout the text. For example, the differential operator in equation (1) and the state in MDP use the same character "s". The transfer function G and the state transition function P should be unified, the current expression is confusing. If a1 and M1 represent the same value, why do the authors use different notations?

5. The control error values in equation (2) are not defined. The error between what? It is suggested that the reference trajectory model be placed in a more appropriate location.

6. What is the difference between the proposed method and TD3?

7. Please improve the quality of all figures and the language.

6. PLOS authors have the option to publish the peer review history of their article (what does this mean?). If published, this will include your full peer review and any attached files.

Reviewer #1: No

---

## [Author Response · Author response to Decision Letter 0]

20 Nov 2023

Dear Editors and Reviewers:

Thanks for your comments concerning our manuscript entitled “Adaptive control for circulating cooling water system using deep reinforcement learning” (ID: PONE-D-23-24165). Your comments are really helpful for revising and improving our paper. We have studied those comments carefully and have made some corrections which we hope to meet with your approval. The main corrections in the paper and the response to the reviewers are as follows:

Comments 1: How does the proposed method ensure the stability of the system?

Response 1: Thank you for your valuable feedback on our submitted paper. We have carefully read your review comments, and in response to your concerns about system stability, we are willing to provide a detailed response.

In this paper, we employ the Twin Delayed Deep Deterministic Policy Gradient (TD3) algorithm designed to improve training stability. By introducing a Twin Q network and a delayed update mechanism, we aim to reduce the variance during training to prevent excessive fluctuations in the system. The algorithm employs experience playback, which is one of the commonly used techniques in the field of deep reinforcement learning. Experience playback helps mitigate instability due to sample correlation and improves the system's robustness by reusing previous experience. In this paper, we rationally design the state space and the reward function of a multivariate system for circulating cooling water to help the deep learning agent perceive the system state more accurately and adjust it according to the reward signal. This initiative aims to prevent unstable behaviours during the training process. In addition, we introduce a reference trajectory model to accelerate convergence and reduce system oscillations during control. This optimization tool helps to make the control system approximate the optimal policy more smoothly and improve the overall stability. 

We encourage regular system performance monitoring in practical applications to enhance safety and system stability further. It is crucial to make adjustments as needed to maintain system stability. Additionally, considering alternative control strategies may be prudent to address specific scenarios that deep reinforcement learning algorithms might not handle effectively. This ensures that the system remains stable even in extreme circumstances.

Based on this, we have made an addition in Section 5. Please refer to the red content in the first paragraph of section 5 on page 13, at line 311~314.

Comments 2: The authors mentioned many successful applications of RL to circulating cooling water system ([26-28]), what is the contribution of this manuscript compared to them? It is suggested that the motivation and contributions should be more emphasized.

Response 2: Thank you for your valuable suggestions on our paper. We understand and value your comments.

Regarding the motivation of the research in this paper, the circulating cooling water system is a complex system with nonlinear, time-lag and multivariate characteristics. Traditional control methods, such as PID controllers, fuzzy control, model predictive control, etc., are often difficult to cope with the complex dynamic characteristics of the system and the uncertainty in the operation process and thus have certain limitations. However, with the development of artificial intelligence technology, reinforcement learning, as a machine learning method based on trial-and-error learning, has powerful nonlinear modelling and adaptive learning capabilities. On the one hand, this paper wants to verify whether deep reinforcement learning has certain advantages over traditional control methods in circulating cooled water systems; on the other hand, although [26-28] have done some research in circulating cooled water-related systems, in general, the research in this field is not deep enough and comprehensive, based on which this paper adopts a different deep reinforcement learning method from [26-28]: the Twin Delayed Deep Deterministic Policy Gradient. The main contributions of this paper are as follows: 1) A deep reinforcement learning controller for circulating cooling water systems was designed based on the TD3 algorithm, achieving end-to-end control and enhancing system stability. 2) The circulating cooling water multivariable system's state space and reward function were reasonably designed. The convergence speed of the agent was accelerated, and the oscillations and instability of the control system were reduced by adding a reference trajectory model. 3) The controller design does not require a model or specialized knowledge about industrial processes. Random disturbance signals were introduced during simulation training to improve the system's adaptive capabilities. 4) The application of deep reinforcement learning in circulating cooling water systems was explored, providing reference and inspiration for control problems in other industrial domains.

Based on this, we have made an addition in Section 1. Please refer to the red content in the third paragraph of section 1 on page 3, at line 69~83.

Comments 3: Since there are many related methods that can also deal with optimal control of unknown systems, it is better to provide a more comprehensive literature review. Please note that the up-to-date of references will contribute to the up-to-date of your manuscript. The studies named: Robust safe reinforcement learning control of unknown continuous-time nonlinear systems with state constraints and disturbances, Journal of Process Control; Online reinforcement learning with passivity-based stabilizing term for real time overhead crane control without knowledge of the system model, Control Engineering Practice, can be used to explain the method in the study or to indicate the contribution in the "Introduction" section. I believe this would further strengthen the introduction and lend support to the methodology used in general.

Response 3: Thank you for your comments and suggestions. The literature you recommended is critical. We fully agree and have added this section to the manuscript.

Please refer to the red content in the third paragraph of section 1 on page 3, at line 61~68.

Comments 4: Check the notation system throughout the text. For example, the differential operator in equation (1) and the state in MDP use the same character "s". The transfer function G and the state transition function P should be unified, the current expression is confusing. If a1 and M1 represent the same value, why do the authors use different notations?

Response 4: Thank you for your comments and suggestions. We apologize for the lack of clarity in our previous presentation; your suggestion is essential.

For this reason, we use "S" to denote the differential operator in Equation (1) and "s" to denote the state in the MDP. The transfer function G is mainly used in control system theory to describe linear time-invariant systems' input and output relationship. In contrast, the state transfer function P is mainly used in the MDP framework to describe the state transfer process between an intelligent body and its environment. In addition, the values of a1 and M1 are indeed the same in this paper, and the reason why different symbols are used is that a1 denotes the value of the action in reinforcement learning, and M1 denotes the value of the valve opening in the circulating cooling water system. The action value a1 obtained after the training of the reinforcement learning algorithm is applied to the circulating cooling water system as a control quantity, and the realization of the control quantity is done through M1.

For the revision details of this question, please refer to the red content in section 2 on page 5, at line 127.

Comments 5: The control error values in equation (2) are not defined. The error between what? It is suggested that the reference trajectory model be placed in a more appropriate location.

Response 5: Thank you for pointing this out. For the revision details of this question, please refer to the red content in section 3 on page 7, at line 177.

The reason for placing the reference trajectory model after the setpoint is that, considering that the setpoint may have sudden changes or instability in practical applications, adding the reference trajectory model after it can smooth out the setpoint signal so that its changes are slower and smoother, thus helping to reduce the oscillations and instability of the control system. The fact that the reference trajectory model is placed after the setpoint does not affect the magnitude of the error value.

Comments 6: What is the difference between the proposed method and TD3?

Response 6: Thank you for pointing this out. We are willing to provide further explanation on the issue.

The control algorithm used in our proposed method is TD3. However, we have adjusted the system's control structure to adapt to the control problems in circulating cooling water systems. Since the setpoints in the circulating water system may have sudden changes or instability in practical applications, this may lead to unstable performance or oscillations in the control system. By introducing a reference trajectory model, the setpoint signal can be smoothed to make its changes slower and smoother, which helps to reduce the oscillation and instability of the control system. From the learning curves of different methods under the same task in Fig. 5, the method proposed in this paper obtains higher rewards faster and more stably due to the addition of the reference trajectory model. Through simulation experiments, the method proposed in this paper has a better performance and a more significant potential in a comprehensive view.

Comments 7: Please improve the quality of all figures and the language.

Response 7: Thank you for your review and valuable comments. We take your suggestions very seriously and have already started to improve the quality of all graphics and language in the paper. We will carefully review and cross-reference your guidance to ensure that all charts and graphs, as well as the presentation of the paper, are more precise, more accurate, and better aligned with academic requirements. We look forward to demonstrating a tangible effort to improve on your suggested improvements in the final version.

Thank you again for your guidance and review.

We are looking forward to hearing from you at your earliest convenience. Thanks for your attention and time.

Sincerely,

Qingxin Zhang (Corresponding author)

E-mail: zhy9712_sau@163.com

Shenyang Aerospace University

November 20, 2023

---

## [Decision Letter · Decision Letter 1]

19 Jan 2024

PONE-D-23-24165R1Adaptive control for circulating cooling water system using deep reinforcement learningPLOS ONE

Dear Dr. Zhang,

Thank you for submitting your manuscript to PLOS ONE. After careful consideration, we feel that it has merit but does not fully meet PLOS ONE’s publication criteria as it currently stands. Therefore, we invite you to submit a revised version of the manuscript that addresses the points raised during the review process.

We look forward to receiving your revised manuscript.

Kind regards,

Lalit Chandra Saikia, PhD

Academic Editor

PLOS ONE

Journal Requirements:

Additional Editor Comments:

All the comments of reviewer must be addressed and necessary changes must be done in the revised manuscript.

Reviewers' comments:

Reviewer's Responses to Questions

**Comments to the Author**

1. If the authors have adequately addressed your comments raised in a previous round of review and you feel that this manuscript is now acceptable for publication, you may indicate that here to bypass the “Comments to the Author” section, enter your conflict of interest statement in the “Confidential to Editor” section, and submit your "Accept" recommendation.

Reviewer #1: All comments have been addressed

Reviewer #2: All comments have been addressed

Reviewer #3: (No Response)

2. Is the manuscript technically sound, and do the data support the conclusions?

Reviewer #1: Yes

Reviewer #2: Yes

Reviewer #3: Yes

3. Has the statistical analysis been performed appropriately and rigorously? 

Reviewer #1: Yes

Reviewer #2: Yes

Reviewer #3: Yes

4. Have the authors made all data underlying the findings in their manuscript fully available?

Reviewer #1: No

Reviewer #2: Yes

Reviewer #3: Yes

5. Is the manuscript presented in an intelligible fashion and written in standard English?

Reviewer #1: Yes

Reviewer #2: Yes

Reviewer #3: Yes

6. Review Comments to the Author

Reviewer #1: The authors have addressed most of my concerns and the paper is recommended for acceptance if possible.

Reviewer #2: (No Response)

Reviewer #3: Some potential drawbacks of the proposed deep reinforcement learning control method:

1. Complexity: Deep RL methods introduce significant complexity compared to traditional controllers.

2. Hyperparameters: Fine-tuning hyperparameters like discount factor, learning rate etc. requires expertise.

3. Sample efficiency: Large volumes of experience/data needed to learn optimal policy, may not be feasible in practice.

4. Brittleness: Policies could fail under distribution shifts or novel operating conditions not seen during training.

5. Non-stationary systems: No mechanism provided to continually learn as system dynamics change over time.

6. Interpretability: Learned policies are black-boxes, hard to analyze causes of behavior and ensure robustness.

7. Real system validation: Only simulated tests conducted, performance on real plant with noises/disturbances unknown.

8. Computational cost: Training deep RL agents is computationally expensive requiring specialized hardware.

9. Data requirements: Need sufficient coverage of state-action space in collected data to train policy.

10. Safety: No fail-safes described for scenarios where control deteriorates before retraining can occur.

11. Single objective: Only optimize for one control metric, may negatively impact other important factors.

12. Keywords section is missing.

13. Describe dataset features in more details and its total size and size of (train/test) as a table.

14. Flowchart and algorithm steps need to be inserted.

15. Time spent need to be measured in the experimental results.

16. Limitation Section need to be inserted.

17. All metrics need to be calculated in the experimental results as tables.

18. Address the accuracy/improvement percentages in the abstract and in the conclusion sections, as well as the significance of these results.

19. The architecture of the proposed model must be provided

20. The authors need to make a clear proofread to avoid grammatical mistakes and typo errors.

21. The authors need to add recent articles in related work and update them.

22. Add future work in last section (conclusion) (if any)

23. Enhance the clarity of the Figures by improving their resolution.

24. To improve the Related Work and Introduction sections authors are recommended to review this highly related research work paper:

a) Building an Effective and Accurate Associative Classifier Based on Support Vector Machine

b) A survey on improving pattern matching algorithms for biological sequences

7. PLOS authors have the option to publish the peer review history of their article (what does this mean?). If published, this will include your full peer review and any attached files.

Reviewer #1: No

Reviewer #2: No

Reviewer #3: **Yes: **Tarek Abd El-Hafeez

---

## [Author Response · Author response to Decision Letter 1]

25 Feb 2024

Response to Reviewer Comments

Thank you very much for taking the time to review this manuscript. We will carefully consider and provide detailed answers to your questions. Please find the detailed responses below and the corresponding revisions/corrections highlighted/in track changes in the resubmitted files.

Comments: Some potential drawbacks of the proposed deep reinforcement learning control method:

1. Complexity: Deep RL methods introduce significant complexity compared to traditional controllers.

2. Hyperparameters: Fine-tuning hyperparameters like discount factor, learning rate etc. requires expertise.

3. Sample efficiency: Large volumes of experience/data needed to learn optimal policy, may not be feasible in practice.

4. Brittleness: Policies could fail under distribution shifts or novel operating conditions not seen during training.

5. Non-stationary systems: No mechanism provided to continually learn as system dynamics change over time.

6. Interpretability: Learned policies are black-boxes, hard to analyze causes of behavior and ensure robustness.

7. Real system validation: Only simulated tests conducted, performance on real plant with noises/disturbances unknown.

8. Computational cost: Training deep RL agents is computationally expensive requiring specialized hardware.

9. Data requirements: Need sufficient coverage of state-action space in collected data to train policy.

10. Safety: No fail-safes described for scenarios where control deteriorates before retraining can occur.

11. Single objective: Only optimize for one control metric, may negatively impact other important factors.

12. Keywords section is missing.

13. Describe dataset features in more details and its total size and size of (train/test) as a table.

14. Flowchart and algorithm steps need to be inserted.

15. Time spent need to be measured in the experimental results.

16. Limitation Section need to be inserted.

17. All metrics need to be calculated in the experimental results as tables.

18. Address the accuracy/improvement percentages in the abstract and in the conclusion sections, as well as the significance of these results.

19. The architecture of the proposed model must be provided

20. The authors need to make a clear proofread to avoid grammatical mistakes and typo errors.

21. The authors need to add recent articles in related work and update them.

22. Add future work in last section (conclusion) (if any)

23. Enhance the clarity of the Figures by improving their resolution.

24. To improve the Related Work and Introduction sections authors are recommended to review this highly related research work paper:

a) Building an Effective and Accurate Associative Classifier Based on Support Vector Machine

b) A survey on improving pattern matching algorithms for biological sequences

Response: 

Thank you for your detailed comments and suggestions on the paper we submitted. We greatly appreciate your interest and contribution to our work. In your suggestions, we recognize that some of these issues deserve further consideration and improvement.

We greatly appreciate your review and valuable feedback on our work. We acknowledge the challenges you raised (issues 1-11) as common concerns faced by deep reinforcement learning methods in the control field. Your insights will help us refine and apply our approach to practical engineering applications. We will carefully consider your suggestions and strive to address these issues in our future work.

Regarding your point about adding a Keywords section (issue 12), we appreciate your suggestion. However, according to the official template and guidelines we followed, a separate Keywords section is optional. For consistency with the format provided by the journal and based on the precedent set by previously published papers on the journal's website, we did not include a Keywords section in our manuscript. Nevertheless, we have provided the relevant keywords for our paper: Industrial process control, Circulating cooling water system, Deep reinforcement learning, PID controller, and TD3.

In response to issue 13 concerning dataset description, we acknowledge your feedback. Our study did not utilize a specific dataset; instead, we conducted research and experiments based on theoretical models and simulation environments. Therefore, we did not provide specific details about a dataset in the manuscript.

As for your issue 14, we are very grateful for your suggestion, which has already been included in the document. Please refer to Figure 4 and Algorithm 1's related content.

Regarding issue 15, we understand your point about time measurement in the experimental results. However, at this stage, we cannot conduct additional time measurements. Therefore, we do not intend to add this information to the manuscript. We will consider incorporating time measurements in future experiments to provide more comprehensive results.

Regarding your issue 16, we appreciate your suggestion. Although our approach achieves good control performance in simulation experiments and shows advantages over both traditional control methods and other deep reinforcement learning methods, we are well aware of some potential limitations, such as applicability constraints, computational resource requirements, hyper-parameter sensitivity, adaptability to environmental variations, and the challenges of practical system validation. Based on this, we have made an addition in Section 5. Please refer to the red content in the second paragraph of section 5 on page 14, at lines 316~320.

Concerning issue 17, we appreciate your recommendation. Performance metrics for controllers, such as rise time, transient time, and overshoot, have been extensively calculated and compared in Table 2. The data presented in Table 2 effectively demonstrate the performance disparities among the controllers and offer readers a comprehensive understanding of the experimental outcomes. If you deem additional performance metrics or further analysis necessary, please suggest, and we will make the necessary adjustments.

For issue 18, we value your suggestion. The manuscript's abstract, experiments, and conclusion sections thoroughly discuss the significance of our research findings. Thanks again for your advice.

Regarding your issue 19, we appreciate your suggestion. Regarding the model of this paper and the system's architecture, which is available in the manuscript, please refer to the relevant part of Figure 4.

Concerning your issue 20 and 23, we take your suggestions very seriously and have already started to improve the quality of all graphics and language in the paper. We will carefully review and cross-reference your guidance to ensure that all charts and graphs, as well as the presentation of the paper, are more precise, more accurate, and better aligned with academic requirements. Thank you again for your guidance and review.

Regarding your issue 21, we appreciate your suggestions. Recent articles on the work studied here are fully cited and discussed in the manuscript, and again, we thank you for your suggestions.

Regarding your issue 22, we appreciate and value your suggestion. Regarding the section on future related work, please refer to the red content in the second paragraph of section 5 on page 14, at lines 320~322.

Regarding your issue 24, thank you very much for reviewing our paper and for the advice you provided. We have carefully considered the references you have given. However, after careful review, they are not directly relevant to the research content of our paper, and therefore, we do not intend to cite them for the time being. We have covered the literature closely related to our research topic in the Related Work and Introduction sections and provided a comprehensive introduction and analysis of related work in the current research area. These references support our paper's background and motivation and give the reader a transparent background and introduction to our research. Please provide specific suggestions if we need to consider citing other relevant literature; we will be more than happy to listen and discuss further.

Thank you again for your valuable suggestions!

---

## [Decision Letter · Decision Letter 2]

2 May 2024

PONE-D-23-24165R2Adaptive control for circulating cooling water system using deep reinforcement learningPLOS ONE

Dear Dr. Zhang,

Thank you for submitting your manuscript to PLOS ONE. After careful consideration, we feel that it has merit but does not fully meet PLOS ONE’s publication criteria as it currently stands. Therefore, we invite you to submit a revised version of the manuscript that addresses the points raised during the review process.

We look forward to receiving your revised manuscript.

Kind regards,

Joanna Tindall

Staff Editor

PLOS ONE

on behalf of: 

Lalit Chandra Saikia

Academic Editor

PLOS ONE

Journal Requirements:

Additional Editor Comments:

The is accepted. All the comments of the reviewers must be addressed.

Comments from Editorial Office: Please address the reviewers comments as outlined by the Academic Editor above under 'Additional Comments'. 

Reviewers' comments:

Reviewer's Responses to Questions

**Comments to the Author**

1. If the authors have adequately addressed your comments raised in a previous round of review and you feel that this manuscript is now acceptable for publication, you may indicate that here to bypass the “Comments to the Author” section, enter your conflict of interest statement in the “Confidential to Editor” section, and submit your "Accept" recommendation.

Reviewer #4: (No Response)

Reviewer #5: All comments have been addressed

Reviewer #6: (No Response)

Reviewer #7: All comments have been addressed

2. Is the manuscript technically sound, and do the data support the conclusions?

Reviewer #4: Partly

Reviewer #5: Yes

Reviewer #6: Yes

Reviewer #7: Yes

3. Has the statistical analysis been performed appropriately and rigorously? 

Reviewer #4: Yes

Reviewer #5: Yes

Reviewer #6: Yes

Reviewer #7: Yes

4. Have the authors made all data underlying the findings in their manuscript fully available?

Reviewer #4: No

Reviewer #5: Yes

Reviewer #6: Yes

Reviewer #7: Yes

5. Is the manuscript presented in an intelligible fashion and written in standard English?

Reviewer #4: Yes

Reviewer #5: Yes

Reviewer #6: No

Reviewer #7: Yes

6. Review Comments to the Author

Reviewer #4: The manuscript proposes a new application of deep reinforcement learning to solve the problem of adaptive control for circulating cooling water systems. The authors have provided a clear explanation of the problem and their proposed solution. Overall, the paper is well-written, and the research question is clearly stated. However, there are some areas that could be improved to enhance the manuscript's clarity and impact.

• The methods section provides a detailed description of the proposed approach, including the deep reinforcement learning algorithm, the simulation environment, and the evaluation metrics. However, it would be helpful to provide more details on the implementation of the algorithm, such as the network architecture, the exploration strategy, and the reward function.

• It would be useful to provide more information on the simulation environment, such as the size and complexity of the system, and how it was validated.

• It would be helpful to provide all metrics in the experimental results as tables.

• I kindly suggest that you address the accuracy and improvement percentages in the abstract and conclusion sections and highlight the significance of these results. This will provide readers with a clear understanding of the impact of your work.

• It would be useful to provide more details on the implementation of the reinforcement learning algorithm, such as the reward function and the network architecture.

• To enhance the manuscript's clarity, I recommend that you add more details about the theoretical models and simulation environments in the experiments section. This will enable readers to better understand the methodology behind your research and potentially replicate your experiments.

Reviewer #5: The paper can be accepted now as the authors have addressed all the comments of the reviewers. The quality of the paper is now overall good.

Reviewer #6: 1. In the abstract part, the method adopted by the author is better than the other 11 control strategies, but the author does not specify the control performance index.

2. There is a syntax error in the introduction, please revise it, between lines 22 and 43.

3. The description between lines 75 and 87 is not appropriate in the introduction, please reconsider.

4. Table 2 should be a three-wire table.

5. The conclusion lacks clarity and should be described objectively..

Reviewer #7: This paper is about the adaptive control for circulating cooling water systems using deep reinforcement learning. There are some issues that the authors have to address:

1. This article aimed to improve the performance of the circulating cooling system. The motivation of this paper should be based on the application. Why is TD3 suitable for the system?

2. What are the differences between the standard TD3 algorithm and the proposed algorithm shown in Algorithm 1? The authors did not provide any details about the improvement.

7. PLOS authors have the option to publish the peer review history of their article (what does this mean?). If published, this will include your full peer review and any attached files.

Reviewer #4: No

Reviewer #5: No

Reviewer #6: No

Reviewer #7: No

---

## [Author Response · Author response to Decision Letter 2]

20 May 2024

Reviewer #4

Comments1: The methods section provides a detailed description of the proposed approach, including the deep reinforcement learning algorithm, the simulation environment, and the evaluation metrics. However, it would be helpful to provide more details on the implementation of the algorithm, such as the network architecture, the exploration strategy, and the reward function.

Response1: Thank you for your valuable comments on our submitted paper. We agree with your suggestions. Please refer to lines 227-237 in Section 3.2 of the manuscript for details on the network architecture of the algorithm. The exploration strategy using Gaussian noise is presented in Table 1. A detailed description of the reward function used in the algorithm can be found in Section 3.1.3.

Comments2: It would be useful to provide more information on the simulation environment, such as the size and complexity of the system, and how it was validated.

Response2: Thank you for your suggestion. In this study, the system is a two-input, two-output system. We trained the control strategy using the TD3-RTM method and then compared it in detail with traditional PID control, fuzzy PID control, DDPG, and the original TD3 algorithm. To ensure fairness and consistency in the comparison, we set the same initial conditions and disturbance factors. Additionally, we used rise time, settling time, overshoot, and IAE metrics to evaluate the performance of the different control systems.

Comments3: It would be helpful to provide all metrics in the experimental results as tables.

Response3: Thank you for your suggestion. All metrics in the experimental results (rise time, settling time, overshoot, and IAE) are presented in tabular form. Please refer to Table 2.

Comments4: I kindly suggest that you address the accuracy and improvement percentages in the abstract and conclusion sections and highlight the significance of these results. This will provide readers with a clear understanding of the impact of your work.

Response4: Thank you for your valuable suggestion. We have revised the abstract and conclusion sections based on your feedback. Please refer to the relevant content in the manuscript.

Comments5: It would be useful to provide more details on the implementation of the reinforcement learning algorithm, such as the reward function and the network architecture.

Response5: As mentioned in your first question, we provided a detailed description of the reward function used in the algorithm in Section 3.1.3. Please refer to lines 227-237 in Section 3.2 of the manuscript for the algorithm's network architecture.

Comments6: To enhance the manuscript's clarity, I recommend that you add more details about the theoretical models and simulation environments in the experiments section. This will enable readers to better understand the methodology behind your research and potentially replicate your experiments.

Response6: Thank you for your feedback. All simulation experiments in this paper were conducted on a PC running Windows 11, equipped with an AMD 4600H CPU @ 3.00 GHz and 16GB of RAM, using MATLAB/Simulink R2022b. The design of the control system (Figure 4) and the settings of algorithm-related hyperparameters have been thoroughly explained in the article. We believe this information will enable readers to better understand the methods behind our research and replicate our experiments.

Reviewer #6

Comments1: In the abstract part, the method adopted by the author is better than the other control strategies, but the author does not specify the control performance index.

Response1: Thank you for your valuable feedback on our submitted paper. We have revised the abstract to provide a detailed description of the control performance metrics, as per your suggestion. Please review the updated abstract section in the manuscript.

Comments2: There is a syntax error in the introduction, please revise it, between lines 22 and 43.

Response2: Thank you for your feedback. The relevant section has been modified accordingly. Please refer to lines 28-50 in the introduction section for the updated content.

Comments3: The description between lines 75 and 87 is not appropriate in the introduction, please reconsider.

Response3: Thank you for your feedback. The relevant section has been modified accordingly. Please refer to lines 76-92 in the introduction section for the updated content.

Comments4: Table 2 should be a three-wire table.

Response4: Thank you for your suggestion. Table 2 has been revised.

Comments5: The conclusion lacks clarity and should be described objectively.

Response5: Thank you for your suggestion. The conclusion section has been revised and objectively described.

Reviewer #7

Comments1: This article aimed to improve the performance of the circulating cooling system. The motivation of this paper should be based on the application. Why is TD3 suitable for the system?

Response1: Thank you for your valuable feedback on our submitted paper. We chose the TD3 algorithm for controlling the circulating cooling water system because it can handle continuous action spaces, utilizes twin Q networks to reduce overestimation bias, and employs delayed policy updates and soft updates to minimize function approximation errors, thereby providing more stable and accurate control. Additionally, TD3's deep neural networks can effectively model the complex nonlinear and multivariate characteristics of the system, enabling real-time adaptation and optimized control, ultimately enhancing system performance. The content has been supplemented in the new manuscript. Please refer to lines 168-174 on page 7 of the manuscript.

Comments2: What are the differences between the standard TD3 algorithm, and the proposed algorithm shown in Algorithm 1? The authors did not provide any details about the improvement.

Response2: The standard TD3 algorithm is not different from Algorithm 1; this study did not modify the TD3 algorithm itself. Instead, it introduced a reference trajectory model to the circulating cooling water system and designed an adaptive control structure for the Twin Delayed Deep Deterministic Policy Gradient algorithm (TD3-RTM) based on this model, as illustrated in Figure 4. The description of this aspect was not sufficiently clear in the original manuscript, but it has been detailed in the newly submitted manuscript.

---

## [Decision Letter · Decision Letter 3]

11 Jul 2024

Adaptive control for circulating cooling water system using deep reinforcement learning

PONE-D-23-24165R3

Dear Dr. Zhang,

We’re pleased to inform you that your manuscript has been judged scientifically suitable for publication and will be formally accepted for publication once it meets all outstanding technical requirements.

Kind regards,

Lalit Chandra Saikia, PhD

Academic Editor

PLOS ONE

Additional Editor Comments (optional):

The paper is recommended for publication.

Reviewers' comments:

Reviewer's Responses to Questions

**Comments to the Author**

1. If the authors have adequately addressed your comments raised in a previous round of review and you feel that this manuscript is now acceptable for publication, you may indicate that here to bypass the “Comments to the Author” section, enter your conflict of interest statement in the “Confidential to Editor” section, and submit your "Accept" recommendation.

Reviewer #6: All comments have been addressed

Reviewer #7: All comments have been addressed

2. Is the manuscript technically sound, and do the data support the conclusions?

Reviewer #6: Yes

Reviewer #7: Yes

3. Has the statistical analysis been performed appropriately and rigorously? 

Reviewer #6: Yes

Reviewer #7: Yes

4. Have the authors made all data underlying the findings in their manuscript fully available?

Reviewer #6: Yes

Reviewer #7: Yes

5. Is the manuscript presented in an intelligible fashion and written in standard English?

Reviewer #6: Yes

Reviewer #7: Yes

6. Review Comments to the Author

Reviewer #6: This paper has clear logic and reasonable structure, has certain innovation and application value, and it is recommended to be published.

Reviewer #7: (No Response)

7. PLOS authors have the option to publish the peer review history of their article (what does this mean?). If published, this will include your full peer review and any attached files.

Reviewer #6: No

Reviewer #7: No

---

## [Editor Report · Acceptance letter]

15 Jul 2024

PONE-D-23-24165R3 

PLOS ONE

Dear Dr. Zhang, 

I'm pleased to inform you that your manuscript has been deemed suitable for publication in PLOS ONE. Congratulations! Your manuscript is now being handed over to our production team.

Kind regards, 

on behalf of

Dr. Lalit Chandra Saikia 

Academic Editor

PLOS ONE